# Epidemiology, Evolution of Antimicrobial Profile and Genomic Fingerprints of *Pseudomonas aeruginosa* before and during COVID-19: Transition from Resistance to Susceptibility

**DOI:** 10.3390/life12122049

**Published:** 2022-12-07

**Authors:** Răzvan Lucian Coșeriu, Camelia Vintilă, Anca Delia Mare, Cristina Nicoleta Ciurea, Radu Ovidiu Togănel, Anca Cighir, Anastasia Simion, Adrian Man

**Affiliations:** 1Department of Microbiology, George Emil Palade University of Medicine, Pharmacy, Science and Technology Târgu Mureș, 540139 Targu Mures, Romania; 2Doctoral School, George Emil Palade University of Medicine, Pharmacy, Science and Technology Târgu Mureș, 540139 Targu Mures, Romania

**Keywords:** *Pseudomonas aeruginosa*, ERIC-PCR, bacterial resistance profiles, COVID-19

## Abstract

Background: The purpose of the study was to describe the epidemiological implication of *Pseudomonas aeruginosa* between 2017–2022 in a tertiary hospital from Romania, including the molecular fingerprinting of similar phenotypic strains (multidrug-resistant isolates), which would have an important health impact. The study also describes the resistance profile of *P. aeruginosa* before and during COVID-19, which might bring new information regarding the management of antibiotic treatments. Materials and methods: Information regarding wards, specimen types, species, and antibiotic resistance profile of 1994 strains of *Pseudomonas* spp. Isolated over a period of 6 years in Mures Clinical County Hospital, Romania, was collected from the WHONET database. From 50 multidrug-resistant isolates, molecular fingerprinting was performed by Enterobacterial Repetitive Intergenic Consensus Polymerase Chain Reaction (ERIC-PCR) to prove the potential clonal distribution. Results: A number of 1994 *Pseudomonas* spp. were isolated between 2017–2022, from which *P. aeruginosa* was the most frequent species, 97.39% (n = 1942). *P. aeruginosa* was most frequently isolated in 2017 (n = 538), with the dermatology department as the main source, mainly from pus secretion. A drop in the harvesting rate was noted in 2020 due to COVID-19 restrictions. Regarding the resistance profile, there are a few modifications. The susceptibility of *P. aeruginosa* to carbapenems, piperacillin-tazobactam, and amikacin suffered alterations before and during COVID-19. The molecular fingerprinting showed three *P. aeruginosa* clusters, including strains with 80–99% similarity.

## 1. Introduction

The increased mortality associated with bacterial infections is partly due to the multitude of resistance mechanisms developed, such as extended-spectrum β-lactamase (ESBL), carbapenem-resistance, methicillin-resistance, and others, which are frequently present in bacteria that cause invasive infections [1]. According to Centers for Disease Control and Prevention (CDC) reports, 2.8 million infections occur with antibiotic-resistant bacteria in the U.S.A. (United States of America), and 35,000 people die each year from one of these infections [2]. Furthermore, the World Health Organization (WHO) stated that infectious diseases kill over 17 million people worldwide each year [3].

An antibiotic prescription is required only when bacteria are proven to be the etiological cause of the infection. By using respiratory infections as an example, antibiotic treatment is overprescribed, even though most cases turn out to be viral [4]. Although the CDC raised awareness of the appropriate use of antibiotics for various types of infections in 1995, misuse of antibiotics is still a major problem in some countries [5]. Even if we talk about outpatient, inpatient, opportunistic or nosocomial infections, statistics show an alarming increase in the numbers around the world [6,7].

There are reports worldwide, not only nowadays, of different resistant bacterial species. There were case reports of antibiotic-resistant *Neisseria gonorrhoeae* in Vietnam in 1967 and later in the Philippines and in the U.S.A. These rarer situations were also found in other types of bacteria, for example, authors who described the penicillin resistance of *Streptococcus pyogenes*, in this case, to the animals. In 2001, there were no infections with carbapenem-resistant *Enterobacterales* (CRE) in the U.S.A, but in 2010, after only 9 years, there were 4% of this type of bacteria, especially *Klebsiella* spp. [8,9]. According to statistics, there were 150,000 patients with *Staphylococcus aureus* MRSA (methicillin-resistant *Staphylococcus aureus*) in the European Union [10]. In Europe, 25,000 people die each year from multidrug-resistant bacteria (MDR) [11]. Analyzing the results of a study carried out in France in the intensive care unit (ICU), from 2015 to 2017, 30% out of a total of 6091 patients hospitalized with pneumonia were diagnosed with infections caused by *S. aureus*, followed by *Pseudomonas aeruginosa* (*P. aeruginosa*) (20.7%) and *Klebsiella* spp. (8.5%), all with varying degrees of resistance. Moreover, 67.1% of patients were diagnosed with a Gram-negative infection during hospitalization [12]. Between 2010 and 2011, 72% of *Klebsiella pneumoniae* with ESBL were isolated in South Africa [13]. In Pakistan, 50 to 60 percent of Gram-negative bacteria isolated from urine are resistant to amoxicillin, cefixime, and ciprofloxacin [14].

All these examples from countries around the world show how quickly bacteria are becoming resistant to antibiotics. Because bacteria adapt so quickly, it can be said that pharmaceutical companies slow down or abandon the development of new antibiotics. Due to the lack of investment, the production of antibiotics is less attractive for pharmaceutical companies [15].

The purpose of our study was to evaluate the involvement of *Pseudomonas* spp. in the infections from our geographical area, as well as to assess the resistance profile and the evolution of antibacterial resistance of these species over the last 6 years, which include before and during COVID-19 timeframes. The applied genetic fingerprinting method provides information about the possible clonal spreading of this species for a better understanding of the infection epidemiology.

## 2. Materials and Methods

Retrospective microbiological reports of Mures Clinical County Hospital (MCCH) from January 2017 to July 2022 were used to analyze the frequency of *Pseudomonas* species. The hospital is divided into 22 clinics, with multiple pathologies, with a large diversity of samples. The WHONET 2021 software was used to centralize and analyze information on *Pseudomonas* strains. As a minimum inclusion criterion, only the first isolates of the strain in the statistical data were selected at this stage, excluding chronic patients with multiple samples. The study was approved by the Ethical Board of MCCH (no. 15190 from 19 October 2020).

*Pseudomonas* spp. was isolated and identified according to the protocols of the microbiology laboratory of MCCH. All the patients’ samples were inoculated on Sheep Blood Agar and Cystine Lactose Electrolyte Deficient (CLED Agar, Oxoid Ltd., Thermo Fisher, Heysham, UK) and incubated at 35 °C, for 16–18 h, according to the routine laboratory protocols. The lactose-negative colonies from CLED Agar were tested for oxidase production and isolated on Cetrimide Agar. Oxidase-positive and cetrimide-positive colonies were considered *P. aeruginosa*. For strains testing negative for cetrimide, the species were identified using the Vitek 2 Compact System (Biomerieux, Marcy-l'Étoile, France) or reported as *Pseudomonas* spp. and included in the statistical data.

All isolates were tested for antimicrobial susceptibility on Mueller Hinton Agar (Oxoid Ltd., Thermo Fisher, Heysham, UK) using the Kirby-Bauer Diffusion method. All tested antibiotics were used according to the European Committee on Antimicrobial Susceptibility Testing (EUCAST) and interpreted based on the data available at the time. The plates were incubated at 35 °C for 16–18 h. The minimum inhibitory concentration (MIC) was tested using the Vitek 2 system (AST N222, Biomerieux, Marcy-l'Étoile, France) for samples with meropenem 10 μg intermediate or resistant. 

The *P. aeruginosa* isolates with intermediate and resistant results to meropenem were stored for further experiments at −70 °C in Tryptic Soy Broth (TSB, Oxoid Ltd., Thermo Fisher, Heysham, UK) containing 10% glycerol.

For the *P. aeruginosa* isolates that presented a high resistance profile, genetic similarity was appreciated using a molecular approach. Thus, from the *P. aeruginosa* stock, we have randomly selected 50 for ERIC-PCR (Enterobacterial Repetitive Intergenic Consensus Polymerase Chain Reaction) from the period 2020–2021, 49 strains from the dermatology department and one from the infectious disease department to detect similarities and appreciate the potential clonal spreading of *P. aeruginosa* in MCCH. ERIC-PCR is based on the amplification of the multiple ERIC palindromic sequences scattered across the bacterial genome, using specific primers that target this region; the reaction concludes in the generation of one or more amplification products of different molecular sizes, which will give a specific electrophoretic pattern for each strain.

The DNA was extracted using Indispin Pathogen Kit (Indical Bioscience, Leipzig, Germany), following the manufacturer’s extraction protocol, and obtaining a final volume of 50 μL of DNA (concentration between 35.6–60.9 µg/µL, with 260 nm/280 nm ratios >1.8).

For ERIC PCR, the reaction mix contained 12.5 μL DreamTaq Green PCR Master Mix 2X (Thermo Fisher Scientific, Waltham, MA, USA), 0.4 mM of Forward ERIC Primer 5′ ATGTAAGCTCCTGGGGATTCAC 3′, 0.4 mM of Reverse ERIC Primer 3′ AAGTAAGTGACTGGGGTGAGCG 5′ (Thermo Fisher Scientific, Waltham, MA, USA), 1μL of DNA, and DNase Free Water to the final volume of 25 μL.

The amplification protocol followed the guidelines presented by Hematzadeh et al. with some modifications, as follows: 5 min initial denaturation at 95 °C, 30 amplification cycles (94 °C for 30 s, annealing 52 °C for 1 min, elongation at 72 °C for 2 min), followed by final elongation at 72 °C for 8 min [16]. *P. aeruginosa* ATCC 27853 was used as the internal control.

Ten microliters of each amplification product were loaded in 2% electrophoresis gel, prepared by mixing 1.4 g of Grade Electran^®^ DNA Agarose with 70 mL of Tris-borate-EDTA (TBE) Buffer. For visualization of DNA, RedSafe^®^ (Sigma-Adrich, Saint Louis, MO, USA) was used for staining. The molecular ladder consisted of 1 μL of Gene Ruler 100 bp (Thermo Fisher Scientific, Waltham, MA, USA), loaded in the first lane of the gel. The electrophoresis was run in 1X TBE Buffer for two and a half hours at 5 Volts/centimeters. The image of the final results was captured using MiniBIS Pro (Bio-Imaging Systems, Modi'in-Maccabim-Re'ut, Israel). The ERIC-PCR dendrogram was created using GelJ Software (UPGMA method, with band matching tolerance established at 1). ERIC patterns reaching 80% similarity were considered identical. *Pseudomonas aeruginosa* ATCC 27853 has been used as control, expecting to form a simple clade.

For statistical data, the results were analyzed using Microsoft Excel and GraphPad Instat 3.

## 3. Results

A retrospective and prospective study were conducted on inpatients and outpatients at the MCCH over a period of five years (January 2017–July 2022). Data from 1994 isolates of the genus *Pseudomonas* were analyzed, including *Pseudomonas aeruginosa*, *Pseudomonas alcaligenes*, *Pseudomonas fluorescens*, *Pseudomonas putida*, *Pseudomonas stutzeri* or non-defined *Pseudomonas* spp.

By analyzing the prevalence of *Pseudomonas* species in MCCH samples, it was shown that *P. aeruginosa* (n = 1942) was the most common isolated species, followed by the rest of the non-aeruginosa species in lesser amounts, such as *Pseudomonas putida* (n = 4) or *Pseudomonas fluorescens* (n = 9). Details on *Pseudomonas* prevalence and demographics are presented in Table 1.

The results showed that of the total number of 1994 *Pseudomonas* isolates, the year 2017 included most of them (n = 547, 21.4%), while the following years identified fewer *Pseudomonas* isolates in terms both of numbers and percentages out of the total number of bacterial isolates in MCCH. In 2018, compared with 2017, the number of isolates strains decreased by 10.66%, despite the fact that the total number of bacterial isolates increased by 62.40%.

As noted in Table 2., comparing the number of *P. aeruginosa* identified each year from all the pathological products with the number of total bacterial isolates using a Chi^2^ and Fisher’s statistical test, it can be concluded that there is a significant increase in the number of *P. aeruginosa* isolates in 2020 and 2021 (OR = 1.23–1.46), compared to the previous years. In 2022, at the end of the COVID-19 pandemic, the prevalence of *P. aeruginosa* among the other bacterial isolates decreased significantly (OR = 0.5–0.77) to a level comparable to the one from pre-COVID-19 years.

The mean age of patients infected with *P. aeruginosa* was 66 years old (DS = 15.56), in a range of less than 1 year (considered newborn) and 98 years. Across all years included in the study, there was no statistical significance number between males 53.81% (n = 1073) compared with females 46.18% (n = 921). The number of males is predominant in all 7 years.

By analyzing the statistical data and for a better understanding of the distribution of samples in MCCH, seven medical groups were followed and presented in Figure 1: ICU (Intensive Care Unit, including the burns department), surgical group (plastic surgery, gastroenterology, obstetrics and gynecology, and orthopedics departments), medical group (nephrology, cardiology, endocrinology, labor medicine, neonatology, and pediatric departments). The HIV department, pneumology, and infectious clinics were added to the infectious diseases group. Dermatology and oncology were separated from the internal medicine group due to the high number of *Pseudomonas* isolates, in the same way as urology was separated out of the surgery group.

The main source of *P. aeruginosa* was dermatology (samples collected mainly from over-infected wounds, erysipelas, calf and foot ulcers) with higher numbers in 2017, 2018, and 2019, mainly from pus secretion 99.36% (n = 783). The number of samples from dermatology drastically decreased in 2020, but this department still remained a major source. In 2021, the number of *P. aeruginosa* isolated from ICU (n = 53) increased by 79.24% compared with 2020 (n = 11), showing a peak after a 3-year lower prevalence; the main sources were represented by tracheal aspirates (n = 27), blood (n = 6) and purulent secretion (n = 8).

Table 3 shows the *Pseudomonas* species identified from 2017 to 2022. Apart from *P. aeruginosa*, only a few other species have been identified, such as *P. alcaligenes*, identified only once in 2022. *Pseudomonas* species were frequently identified in purulent secretions referred from the dermatology department, which was the main source each year.

In 2020, due to the pandemic situation of COVID-19, some hospital wards were reassigned to support COVID-19 case management. Consequently, the number of chronic pathologies, thus the number of common bacteriological tests, was reduced. The total number of *P. aeruginosa* identified in 2020 was only n = 117, which represents a 78.76% decrease compared with the previous year. Nevertheless, the purulent secretions remained the main source of *P. aeruginosa*.

Studying the resistance profile of *P. aeruginosa* isolated from 2017–2022, it can be observed that it did not modify considerably, with a few exceptions. Figure 2 presents the susceptibility profiles for representative antibiotics used according to EUCAST, and the number of *P. aeruginosa* resistant (R), susceptible (S), and susceptible, increased exposure (I). Due to a series of limiting factors, such as adjustments in the EUCAST standard interpretation data, and situations when not all the antibiotics were tested for all isolates, the number of the results presented in the resistance profile of *P. aeruginosa* (Figure 2) might differ from the total number of isolated *P. aeruginosa*.

All selected strains for ERIC-PCR fingerprinting presented pandrug-resistance to antibiotics. Thus, genetic testing is important to prove the genetic relation between *P. aeruginosa* strains in MCCH. The majority of *P. aeruginosa* selected for testing were isolated from the department of dermatology (92.5%) and one of the strains from the infectious disease department, considering the high prevalence of *P. aeruginosa* isolates from this ward, from pus secretion.

Electrophoresis of ERIC-PCR products showed the definition of several amplicon bands in 42 of the 50 tested strains, with molecular weights ranging from 300 bp to 1500 bp; eight strains were non-typeable and were excluded from the analysis. By analyzing the dendrogram results (the threshold for potentially similar strains set at 80%), a number of three clusters were defined, indicating a possible ancestral relationship from the isolates (Figure 3). Completely identical ERIC-PCR patterns were not identified within the studied isolates; nevertheless, similarity >98% was observed in a few cases.

The first cluster was formed by seven strains, which were isolated from pus secretion from samples in the department of dermatology, with a grade of similarity of 81%. The finding is supported by the fact that the patients were admitted to the clinic in 3 consecutive months (January 2021 n = 2, February 2021 n = 2, and March 2021 n = 3) at a difference of a few days to one week. What is more interesting, four of them followed treatment with ceftriaxone. The cumulative antibiogram of these isolates showed pan-resistance, with sensitivity retained to colistin. The strains 16, 18, and 19, 21 had a similarity of 98–99%.

The second cluster included a higher number of strains (n = 11) isolated from patients hospitalized in February 2021 (n = 3), April 2021 (n = 4), June 2021 (n = 2), and July 2021 (n = 2). The patients were admitted to the dermatology department (n = 10) and infectious diseases clinic (n = 1) without a history of transfer among departments. Samples 32, 45, and 24, 25 showed a similarity of 98–99%.

The third cluster was formed by eight strains of *P. aeruginosa* from patients admitted to the dermatology department in a period of one year (2021), without a specific period (January, July, August, and September 2021).

## 4. Discussion

Even if the number of *Pseudomonas* spp., compared to other bacterial species isolated in MCCH, represented only a quarter of the total isolates, it is considered a challenge in the process of keeping under control and treating the patients [17]. It has been established that *P. aeruginosa* is one of the species from the ESKAPE group, an acronym that describes the most frequently encountered nosocomial bacteria: *Enterococcus faecium*, *Staphylococcus aureus*, *Klebsiella pneumoniae*, *Acinetobacter baumannii*, *Pseudomonas aeruginosa,* and *Enterobacter* spp. All these species can present high antibiotic resistance and can easily prevail in the hospital environment, thus representing a high epidemiological risk [18]. The resistance profile of *Pseudomonas* bacteria can be influenced by the environmental surroundings or by the treatment of the patients, while its aggressiveness is by the numerous virulence factors that can be produced; thus, due to the high adaptability, *Pseudomonas* can be considered a survivor bacteria [19]. Other studies found *P. aeruginosa* as the most frequent isolate from all analyzed pathological products [20,21,22], compared to the data from our study where *P. aeruginosa* seconds *S. aureus.*


A report issued in the U.S.A. by CDC explains the data about *P. aeruginosa*, and what has to be emphasized is the fact that the number of MDR *P. aeruginosa* decreased from 2011 to 2022 to less than 8% [23]. In our opinion, the declining number of *P. aeruginosa* can be explained by underreporting of MDR *P. aeruginosa* by some laboratories; this can be seen in the 2020 European Center for Disease and Control (ECDC) report, which included 29 countries, but only 675 *P. aeruginosa* isolates have been reported [24]. In our hospital alone, 117 *P. aeruginosa* strains were isolated in 2020. ECDC published a document showing all the European countries that reported *P. aeruginosa* and classified it in fourth place as the most common hospital-related pathogen, representing 9%. The same ECDC publishes periodical data from all European countries that report *P. aeruginosa* infections, including Romania, a statistic which, in our opinion, is underestimated. For example, according to ECDC, Romania reported in 2017 a number of 132 strains of *P. aeruginosa* from ICU [24], despite the fact that, according to our study data, 84 (63.63%) strains of *P. aeruginosa* were isolated in our hospital in ICU alone. In the USA, according to the CDC, the *P. aeruginosa* infection rate is only 7%, but compared to other studies, there were reported 28%, which only reveals an underestimation of the numbers. In India, Iran, Pakistan, and Nigeria, there are studies that show the isolation of *P. aeruginosa* in a range between 6.67–30% [25,26,27,28].

By analyzing the studies regarding the distribution of *P. aeruginosa*, it can be concluded that it does not have a preference for a specific infection site. Thus, the pathological products are variate. For example, in older studies conducted during 2001–2003, in Bangladesh and Thailand, it was shown that the most frequent pathological product with isolation of *P. aeruginosa* was sputum, contradictory with our results, where pus secretion was definitely the most frequent. Our results showed that LRT secretions (sputum and tracheal aspirate) represented between 2.7–11% (differing from year to year) of the pathological products, with *P. aeruginosa* identification, the first and second place being occupied by pus secretion and urine [29]. A study conducted in Saudi Arabia in 2014–2015 also concluded that *P. aeruginosa* was identified mostly from sputum, while a study conducted for one year in 2020 in Egypt placed blood culture as the main source for *P. aeruginosa*, followed by urine and pus secretion from wounds [30,31]. All the above examples only support the idea that *P. aeruginosa* can be identified from a wide variety of infectious sites with different frequency distributions.

Some authors consider *P. aeruginosa* to be a superbug [21], but according to our results, the strains are able to transform in time to become superbugs. As can be seen from our data, the antibiotic resistance profile modifies over a period of time, even if the isolated strains are resistant to the majority of antibiotics. The molecular structure of *P. aeruginosa* does not help either. According to EUCAST “Expected Resistance Phenotypes”, all the strains have intrinsic resistance to some class of antibiotics, and for that, the treatment became more difficult. By consulting the EUCAST tables, it can be noticed that *P. aeruginosa* presents resistance to ampicillin, amoxicillin-clavulanate, cefotaxime, ertapenem, and other antibiotics, reducing the chance of treating the infection [32]. More than that, Bothelho et al. considered that during 2011–2015, important changes occurred in the resistance profile of *P. aeruginosa*, highlighting the increased prevalence of carbapenemase or β-lactamase producing strains (which represented, according to ECDC, 27.7%), but also the presence of other mechanisms of resistance, including resistance to colistin; all these rises further challenge in the treatment options in the ICU [33].

What is unexpected is the year 2020, when the number of samples and of isolated *P. aeruginosa* (5.64%) drastically dropped due to the COVID-19 pandemic situation. By searching the Pubmed database using terms [“COVID” AND “Pseudomonas” AND “resistance”], no relevant results were found. This makes our study the first to show the difference in frequency and distribution of *P. aeruginosa* during the COVID-19 pandemic. Even if the number of admitted patients in the ICU was higher during COVID-19, the number of *P. aeruginosa* infections in this department was lower than in 2019. More than that, the resistance profile of *P. aeruginosa* has also changed. The number of carbapenem-resistant strains decreased compared to 2019; also, susceptibility to amikacin was improved. Due to the modification of the structure of our hospital, some wards were redesignated for COVID-19 support. Thus, the main source of *P. aeruginosa* diminished. The chronic patients had limited access to medical care, especially during isolation situations and lockdowns. Hospitalizations were allowed only in emergency situations. The presented situation raises important questions: how did the patients with *P. aeruginosa* infections manage their situation if they did not receive treatment? Can this be another proof that *P. aeruginosa* causes colonization with no necessary treatment? There are authors who consider that the presence of *P. aeruginosa*, especially in wounds or pus secretion, represents, in most cases, colonization, along with other bacteria [34]. Other authors consider that colonization is one of the first steps of infection, followed by dissemination in the bloodstream and causing septicemia [35]. A study conducted in Hong Kong on 1066 patients with non-cystic fibrosis bronchiectasis showed that in 27% of cases, *P. aeruginosa* was a colonizer, but nevertheless, the colonization was associated with a worse prognosis [36]. Other studies also present the importance of the respiratory tract colonization Kunadharaju et al. showed that from more than 22,000 patients with chronic obstructive pulmonary disease, 4.2% presented repeated *P. aeruginosa* isolation from sputum for as long as 36 months [37]. In our results, the number of *P. aeruginosa* strains isolated during the pandemic situation, especially from the dermatology department (the main source in our situation), dropped down from n = 361 (12.31%) in 2019 to n = 81 (10.14%) in 2020; a high number of patients “disappeared” in 2020, due to COVID-19 restrictions.

The genetic investigation of the nosocomial infections caused by *P. aeruginosa* is important in the epidemiological context, especially when the phenotype discrimination is not relevant [38]. Moreover, genetic fingerprinting is important to find the source of a potential outbreak. For example, a study in Iran assessed the relationship between *P. aeruginosa* isolates from the hospital and environmental sources, including cockroaches, by ERIC-PCR and found five distinct clusters with a cutoff of 95% similarity [39]. Our study has identified three main clusters with a similarity of >80%, but very few isolates were completely identical. The situation of non-typeable strains (as in the case of our eight isolates that showed no bands following ERIC-PCR) was also described in other similar studies [40,41].

Many studies conducted in the previous years focused mostly on cystic fibrosis patients, also showing genetic diversity in hospital environments. For example, Syrmis et al. analyzed the similarity of 163 strains of *P. aeruginosa* isolated from patients with cystic fibrosis, with a similarity threshold of 85%, and identified six major clusters and 58 distinct clonal groups [42]. A review from 2018 assessed the global prevalence of *P. aeruginosa* clonality by ERIC-PCR, PFGE, or MLST, but in cystic fibrosis patients; the data were reported by a limited number of countries, but nevertheless showed the sharing of *P. aeruginosa* strains among individuals with this specific pathology [43]. A study in the Netherlands presented a 3-year clonal outbreak of carbapenem-resistant *P. aeruginosa* infections, which was unnoticed until molecular testing was performed [44]. This further highlights the importance of epidemiological surveillance by genotypic methods, complementary to the phenotypical findings.

Identical genetic patterns are suggestive of the clonal distribution of bacteria and infection outbreaks. This was presented, for example, by Opperman et al. in 2022, who found that the ST303 clone was spreading throughout the city, affecting especially the neonates [45]. Another outbreak was proved by molecular methods by Bertrand et al. in 2000 in France, when 55 identical carbapenem-resistant *P. aeruginosa* were isolated in a 30-month time frame from an ICU unit [46]. The clonal spreading is not our case, as the ERIC-PCR patterns were, with few exceptions, not identical, even if the samples originated from the same medical ward and from the same time frame.

In Romania, there have been policies regarding the surveillance, prevention, and limitation of infections associated with medical assistance in health facilities since 2016, which guides the measures to be taken to limit this phenomenon [47]. Despite this, the antibiotic resistance rate was and is still high in Romania. Nevertheless, our data show that in the last years, probably under the pressure of the COVID-19 pandemic that forced medical professionals to better comply with antibiotic surveillance or due to the dissemination of antibiotic therapy guides, the resistance patterns changed to a more favorable situation. This is also shown by the European Centre for Disease Prevention and Control [48] as the percentage of carbapenem- or fluoroquinolone-resistant *P. aeruginosa* was lower in 2020–2021 compared with 2018–2019.

## 5. Conclusions

The data analyzed during 2017–2022 show that *Pseudomonas aeruginosa* is the most prevalent species of the *Pseudomonas* genus, mainly isolated from pus secretion.

The number of *P. aeruginosa* isolates decreased significantly during the COVID-19 pandemic, but at the same time, its prevalence among other bacterial species increased significantly. This can be related to the low addressability to medical services, especially for chronic patients, thus a lower chance to isolate and treat the more “classical” pathogens. Associated with this situation, we show that the resistance profile of *P. aeruginosa*, mainly against carbapenems, fluoroquinolones, and aminoglycosides, was improved in 2020–2022, probably to the limitation of antibiotic abuse in hospitals. The dermatology ward presents the highest risk of selecting and spreading antibiotic-resistant *P. aeruginosa*, many of them being found, by molecular fingerprinting, to be closely related.

All these prove the high adaptability of *P. aeruginosa* in relation to changes in medical protocols, patient addressability, or treated pathologies. This species has the ability to transform easily into a superbug.

## Figures and Tables

**Figure 1 life-12-02049-f001:**
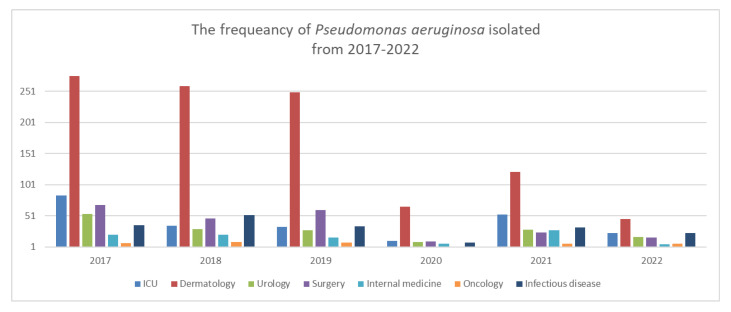
The frequency of *P. aeruginosa* between 2017–2022, in MCCH departments, by groups.

**Figure 2 life-12-02049-f002:**
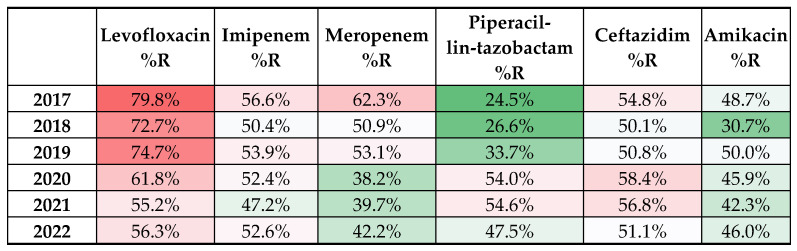
Heatmap presenting the dynamics of resistance profile of *P. aeruginosa* regarding the main antibiotics used in treatment. The color gradient from red to green is representative for higher, respectively lower percentages of resistant strains.

**Figure 3 life-12-02049-f003:**
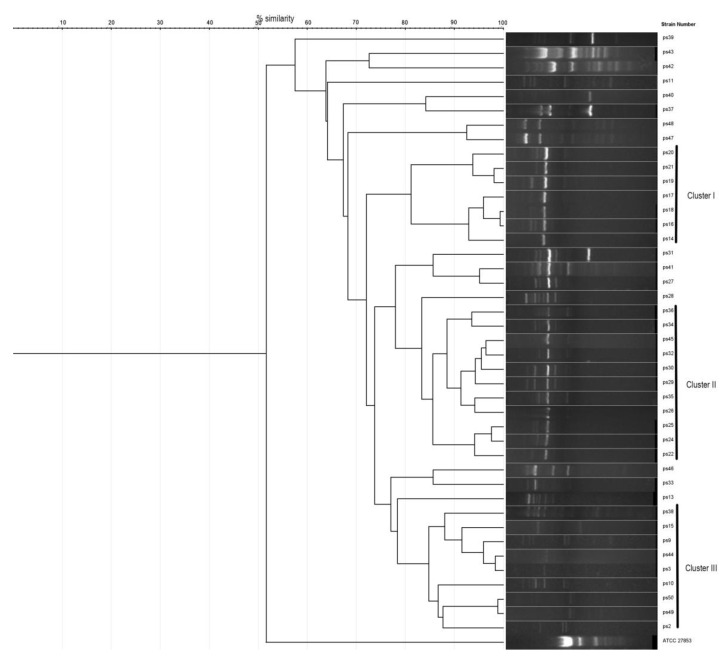
Strains 1–42: ERIC-PCR patterns of *P. aeruginosa* isolates; ATCC 27853: positive control *Pseudomonas aeruginosa*.

**Table 1 life-12-02049-t001:** The distribution of *Pseudomonas* spp. between 2017–2022.

Year	Total Number of Bacterial Isolates in MCCH	Total Number of *Pseudomonas* spp. Isolated (n, %)	Males (n, %)	Females (n, %)	Average Age (Years) (SD-Standard Deviation)
2017	2556	547 (21.4%)	289 (52.83%)	258 (47.17%)	68 (SD = 14.1)
2018	4243	456 (10.74%)	252 (55.26%)	204 (44.74%)	67 (SD = 15.65)
2019	3559	434 (12.19%)	223 (51.38%)	211 (48.62%)	66 (SD = 16.21)
2020	658	117 (17.78%)	66 (56.41%)	51 (43.59%)	65 (SD = 17.33)
2021	1879	299 (15.91%)	163 (54.51%)	136 (45.49%)	66 (SD = 15.68)
2022	1509	141 (9.43%)	80 (56.73%)	61 (43.27%)	65 (SD = 16.19)

**Table 2 life-12-02049-t002:** Statistical data presenting the Odds Ratio (OR) and *p*-values obtained after comparing the number of *P. aeruginosa* isolated each year with the total number of bacterial isolates. Values marked in bold represent statistical significance.

Year	2018	2019	2020	2021	2022
2017	*p* = 0.82 OR: 0.93	*p* = 0.43 OR: 0.94	* **p** * **= 0.0041** **OR: 1.37**	***p*** < **0.0001** **OR: 1.23**	***p*** = **0.0013** **OR: 0.72**
2018	-	*p* = 0.94 OR: 1.00	***p*** = **0.0007** **OR: 1.46**	***p*** = **0.0007** **OR: 1.31**	***p*** = **0.01** **OR: 0.77**
2019	-	-	***p*** = **0.0009** **OR: 1.45**	***p*** = **0.001** **OR: 1.3**	***p*** = **0.01** **OR: 0.76**
2020	-	-	-	*p* = 0.37 OR: 0.52	***p*** < **0.0001** **OR: 0.52**
2021	-	-	-	-	***p*** < **0.001** **OR: 0.58**

**Table 3 life-12-02049-t003:** A summary of statistical data results between 2017–2022.

Number of *Pseudomonas* Isolates (n = 1994) from 2017–2022							
Species	*P. aeruginosa*	*P. alcaligenes*	*P. fluorescens*	*P. putida*	*P. stutzerii*	*Pseudomonas* spp.	
2017	98.35% (n = 538)	-	0.18% (n = 1)	0.18% (n = 1)	-	1.27% (n = 7)	
2018	98.02% (n = 447)	-	0.21% (n = 1)	-	-	1.75% (n = 8)	
2019	96.08% (n = 417)	-	0.46% (n = 2)	-	-	3.45% (n = 15)	
2020	100% (n = 117)	-	-	-	-	-	
2021	97.99% (n = 293)	-	0.66% (n = 2)	1.00% (n = 3)	-	0.33% (n = 1)	
2022	92.19 (n = 130)	0.7% (n = 1)	2.12% (n = 3)	1.41% (n = 2)	3.54% (n = 5)	-	
**Department**							
**Years**	**2017**	**2018**	**2019**	**2020**	**2021**	**2022**	
ICU	15.35% (n = 84)	7.67% (n = 35)	7.6% (n = 33)	9.4% (n = 11)	17.72% (n = 53)	16.31% (n = 23)	
Surgical	12.43% (n = 68)	10.3% (n = 47)	13.82% (n = 60)	0.85% (n = 10)	8.02% (n = 24)	11.34% (n = 16)	
Medical	3.83% (n = 21)	4.6% (n = 21)	3.68% (n = 16)	5.12% (n = 6)	9.36% (n = 28)	3.54% (n = 5)	
Infectious diseases	6.58% (n = 36)	11.40% (n = 52)	7.83% (n = 34)	6.83% (n = 8)	10.7% (n = 32)	16.31% (n = 23)	
Dermatology	50.63% (n = 277)	57.45% (n = 262)	58.75% (n = 255)	60.68% (n = 71)	42.47% (n = 127)	36.17% (n = 51)	
Oncology	1.27% (n = 7)	1.97% (n = 9)	1.84% (n = 8)	1.7% (n = 2)	2.00% (n = 6)	4.25% (n = 6)	
Urology	9.87% (n = 54)	6.57% (n = 30)	6.45% (n = 28)	7.69% (n = 9)	9.69% (n = 29)	12.05% (n = 17)	
**Infection site/Pathological products**							
	**2017**	**2018**	**2019**	**2020**	**2021**	**2022**	**Total**
LRT ^1^	8.75% (n = 48)	9.21% (n = 42)	2.76% (n = 12)	11.11% (n = 13)	16.72% (n = 50)	4.96% (n = 7)	172
Pus	74.77% (n = 409)	74.56% (n = 342)	83.17% (n = 361)	76.06% (n = 89)	60.86% (n = 182)	60.28% (n = 85)	1468
Urine	12.61% (n = 69)	12.93% (n = 59)	9.9% (n = 43)	11.11% (n = 13)	15.38% (n = 46)	21.27% (n = 30)	260
Blood	0.91% (n = 5)	0.87% (n = 4)	0.46% (n = 2)	0.85% (n = 1)	2.00% (n = 6)	1.41% (n = 2)	20
Others	2.92% (n = 16)	1.97% (n = 9)	3.68% (n = 16)	0.85 %(n = 1)	5.01% (n = 15)	12.05% (n = 17)	74

^1^ LRT-Lower Respiratory Tract; Others (bile, cervical secretion, middle and outer year secretion, urethral secretion, synovial fluid, pleural fluid, catheter).

## Data Availability

Not applicable.

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
