# Peer review of "Epidemiology, Evolution of Antimicrobial Profile and Genomic Fingerprints of Pseudomonas aeruginosa before and during COVID-19: Transition from Resistance to Susceptibility"

_life, 2022, doi:10.3390/life12122049_

Round 1
Reviewer 1 Report
In this manuscript, the epidemiological and genomic evolution and resistance to antibiotics of Pseudomonas aeuroginosa were determined during five years.
The following comments are made:
1. Do not use the abbreviation Pae, the scientific abbreviation is P. aeruginosa. Correct throughout the text
2. Line 79. “during-and post-COVID-19 timeframes”. We are not yet in the post-COVID-19 stage, the WHO has not declared the end of the pandemic.
3. Line 116. “For ERIC PCR”. Enter the concentrations used, not the mL.
4. What region are the primers used in ERIC PCR from? Indicate it. How many amplicons should be formed and what is the size of the expected amplicons? Explain it.
5. Line 125. “we used Pseudomonas aeruginosa ATCC 27853”. Put the scientific name of the bacterium in italics. Review and correct throughout the text.
6. Line 136. Describe what type of patients the samples were taken from, what type of illness they had, where the samples were taken from.
7. Line 155 and Table 2. Put what OR means.
8. Line 156. “at the end of the COVID-19 pandemic”. The WHO has not considered the pandemic over.
9. Line 162. Put what DS means
10. Figure 2. Put the results in percentages
11. Line 208. “with variable susceptibility to colistin.” These values ​​are not in Figure 2.
12. Line 211. “8 samples did not present amplification”. Why don't they show amplification? Discuss it
13. Figure 3. Show the similarity percentage scale.
14. Figure 3. There is some clonal relationship of the strains with respect to the site of isolation or the disease. Discuss it.
15. Line 227. “The strains 16, 27 and 19, 20 had a similarity of 98-99%.” This result is not observed in Figure 3
16. Line 232. “Sample 34, 45 and 24, 232 25 showed a similarity of 98-99%”. The result of strains 34 and 35 is not observed in Figure 3.
17. Line 251. “Pae the most frequent cause of infections”. What kind or where? explain it
18. Line 337. “clonality by ERIC-PCR, PFGE or MLST”. How similar are the clonality analyzes between ERIC-PCR and PFGE or MLST? Can they give similar results? Discuss it
19. Line 353. “mainly isolated from pus secretion”. From what kind of disease? explain it
Author Response
We ensure you of our appreciation for the time spend trying to help us improve the quality of the manuscript! The reviewers’ comments were highly constructive and we are totally agreeing that they helped improving the manuscript. Hoping that our answers will respond to your expectations, we are addressing your comments point-by-point.
Reviewer#1
- Do not use the abbreviation Pae, the scientific abbreviation is P. aeruginosa. Correct throughout the text
Thank you! Indeed, is more academic using P. aeruginosa form.
- Line 79. “during-and post-COVID-19 timeframes”. We are not yet in the post-COVID-19 stage, the WHO has not declared the end of the pandemic.
Thank you, we corrected
- Line 116. “For ERIC PCR”. Enter the concentrations used, not the mL.
Thank you! We added the range of concentration that we obtained
- What region are the primers used in ERIC PCR from? Indicate it. How many amplicons should be formed and what is the size of the expected amplicons? Explain it.
Thank you, we explained the ERIC-PCR principle as follows: “ERIC-PCR is based on the amplification of the multiple ERIC palindromic sequences scattered across the bacterial genome, using specific primers that target this region; the reaction concludes in generation of one or more amplification products of different molecular sizes, which will give a specific electrophoretic pattern for each strain.”
- Line 125. “we used Pseudomonas aeruginosa ATCC 27853”. Put the scientific name of the bacterium in italics. Review and correct throughout the text.
Thank you! We changed the name of bacteria in italics
- Line 136. Describe what type of patients the samples were taken from, what type of illness they had, where the samples were taken from.
Thank you! Indeed, it must be explained, the hospital has 22 departments with all specialties. The principal pathological products are presented in Table 3. at section “Infection site / Pathological products”, we described in Material and methods, and we added the main illness
- Line 155 and Table 2. Put what OR means.
Thank you, we explained it in the table 2 description
- Line 156. “at the end of the COVID-19 pandemic”. The WHO has not considered the pandemic over.
Thank you, indeed that’s right, we modified
- Line 162. Put what DS means
Indeed, it must be explained, thank you.
- Figure 2. Put the results in percentages
As the Reviewer#3 recommended, we changed the graphs into a heatmap of the 6 antibiotics, to see more clearly the evolution of the antibiotic resistance. Initially it was represented by graphics to see the evolution also as numbers in time, the percents being described in the text for the most important results.
- Line 208. “with variable susceptibility to colistin.” These values ​​are not in Figure 2.
According to EUCAST, colistin has to be tested by microdilution method. The Vitek or disk-diffusion methods that we used are not acceptable anymore according to the new standards (even if we screened this by the previous standards). Thus, we choose not to include this information in the article.
- Line 211. “8 samples did not present amplification”. Why don't they show amplification? Discuss it
Thank you! We explained it, in discussion section
- Figure 3. Show the similarity percentage scale.
Thank you, we explained it in the Figure
- Figure 3. There is some clonal relationship of the strains with respect to the site of isolation or the disease. Discuss it.
Thank you for the recommendation. We updated the results section.
- Line 227. “The strains 16, 27 and 19, 20 had a similarity of 98-99%.” This result is not observed in Figure 3
Thank you we corrected!
- Line 232. “Sample 34, 45 and 24, 232 25 showed a similarity of 98-99%”. The result of strains 34 and 35 is not observed in Figure 3.
Thank you for noticing this, it was a mistake, we corrected
- Line 251. “Pae the most frequent cause of infections”. What kind or where? explain it
“As described in other publications, some authors consider P. aeruginosa the most frequent cause of infections. In our study (data not shown), P. aeruginosa occupied the second place after S. aureus, this being shown also in other studies”
was repleaced with
„Other studies found P. aeruginosa as the most frequent isolate from all analyzed pathological products, compared to the data from our study where P. aeruginosa seconds S. aureus (data not shown).”
- Line 337. “clonality by ERIC-PCR, PFGE or MLST”. How similar are the clonality analyzes between ERIC-PCR and PFGE or MLST? Can they give similar results? Discuss it
Indeed, there are other methods than ERIC that can assess the similarity. Though, ERIC-PCR is described as a very convenient screening method for genetic similarity. Moreover, we have successfully used the method in previous studies: https://jidc.org/index.php/journal/article/view/35656954
- Line 353. “mainly isolated from pus secretion”. From what kind of disease? explain it
Thank you form the observation! The diseases and the illness were presented.

Reviewer 2 Report
A manuscript is submitted on the topic of epidemiology, evolution of antimicrobial profile and genomic fingerprints of Pseudomonas aeruginosa before and during Covid-19. Undoubtedly, this is a current and beneficial topic.
Unfortunately, the manuscript contains a number of inaccuracies and in its current form I cannot recommend it for acceptance.
I have the following comments on the submitted manuscript, which are listed in a separate file.

Author Response
We ensure you of our appreciation for the time spend trying to help us improve the quality of the manuscript! The reviewers’ comments were highly constructive and we are totally agreeing that they helped improving the manuscript. Hoping that our answers will respond to your expectations, we are addressing your comments point-by-point.
Reviewer #2
- The total number of Pae vs. Pseudomonas isolates
Row 142: „shown that Pae (n=2024) was the most common“ does not correspond with
Row 147: „ The results showed that of the total number of 2076 Pae isolates.“
The total number does not fit. The author apperantly meant: „ The results showed that of the total number of 2024 Pae isolates.“
Thank you, indeed it was a mistake, we have rechecked all the data and corrected.
- Row 148: The results showed that of the total number of 2076 Pae isolates, the year of
2017 147 (n=547, 21.4%) included most of them.”
The sentence should be rewritten to clarify the numbers and percentage.
Thank you, we detailed the percentage to be clearer.
- Table 1: The total number of Pae isolates according to the table is 1994, not 2024.
Thank you, indeed it was a mistake, we have rechecked all the data and corrected it.
- Row 148: “while 2020 was the opposite year, with the smallest number of isolated Pae”.
But the percentage of isolated Pae for year 2020 is not the smallest (17,78 %). Should not we look at the percentage rather at the numbers of isolates? In this case, the lowest prevalence of Pae was during the year 2022 (9,43%) – but of course, the end of this year is not included.
Thank you. Indeed, the rate of P. aeruginosa differed and should be considered aside the actual numbers. We changed the phrase accordingly.
- Materials and methods – when it is described the dendrogram creation based on the ERIC
patterns, there always should be the information about the optimization and band
matching tolerance (did you set up the tolerance at 1,5 or 1?).
Also there should be a note about the % similarity at which you consider the isolates identical e.g. „ERIC patterns reaching x % similarity were considered identical.“
Thank you for the kind suggestion. Indeed, the similarity threshold was mentioned in result only. We also updated the dendrogram methodology as suggested.
- The total number of Pae isolates was 2024 but for the ERIC PCR analysis were randomly chosen 50 isolates from the years 2020 and 2021 (it means from the approx. 416 isolates gained during the years 2020 and 2021 were chosen 50). Be more specific, how many isolates from which department? And why did you choose only 50?
There is still 366 of Pae isolates for the years 2020 and 2021 which were not analysed by any genotyping method. I would not claim the result for 50 isolates out of 2024 is appropriate. I would rather claim we could have gain just an idea if the isolates were clonal.
Thank you, we clarified this in methods: “For the P. aeruginosa isolates that presented a high resistance profile, genetic similarity was appreciated using molecular approach. Thus, from the P. aeruginosa stock, we have randomly selected 50 for ERIC-PCR (Enterobacterial Repetitive Intergenic Consensus Polymerase Chain Reaction), to detect similarities and to appreciate the potential clonal spreading of P. aeruginosa in MCCH”. The results present: “The majority of P. aeruginosa selected for testing were isolated from the department of dermatology (92.5%), considering the high prevalence of P. aeruginosa isolates from this ward, from pus secretion”
- According to the data in the Hematzadeh publication (reference 16), the RAPD PCR with
272 primer provides higher discriminatory power than ERIC PCR. Have you considered the RAPD PCR analysis instead of ERIC PCR? Why did you choose ERIC PCR for genotyping the isolates? It would be also better to combine the ERIC PCR results with some other genotyping method, e.g. RAPD or PFGE, so you will be able to compare it and then make some conclusion.
Indeed, there are other methods than ERIC that can assess the similarity. Though, ERIC-PCR is described as a very convenient screening method for genetic similarity. Moreover, we have successfully used the method in previous studies: https://jidc.org/index.php/journal/article/view/35656954
- Figure 3: the axes should be described – what do we observe at horizontal and vertical-
axis? What represents the scale at the top of dendrogram?
Thank you for noticing, we described the axes.
- Figure 3: The scale at the top of dendrogram (I guess it is the percentage of similarity) and also the ID of isolates should be readable.
Thank you, we updated figure 3 with “% similarity” and “Strain number”
- Row 227: “The strains 16, 27 and 19,20 had a similarity of 98-99%“
According to the scale at the top of the dendrogram, isolates 16 and 27 does not show 98-99% similarity. Neither isolates 19 and 20. According to the created dendrogram, the 98-99% similarity has e.g. isolates 19 and 21 - although it looks like that isolates 20 and 21 are even more related judging just from the ERIC patterns shown in dendrogram.
In general, it seems that ERIC patterns do not correspond with the created dendrogram. I recommend recreation of phylogenetic tree with different set of band matching tolerance.
Thank you for noticing! Indeed it was a mistake of passing the isolate numbers in the text
- Row: 213 “..expecting to form a simplicifolious clade.” I suggest rewriting it “.expecting to form a single clade.”
Thank you, changed!
- Row 232: “Sample 34, 45 and 24,25 showed a similarity of 98-99%” Are you sure about the 34 and 45? But again, recreation of dendrogram would be appropriate.
Thank you for noticing! We have revised the dendrogram data and naming. We fixed the discrepancies in the text.
- Row 233-234: “Representation of dendrogram revealed a similarity of 70% of cluster 1 and 2, indicating a possible ancestral relationship from the isolates” What about the cluster 3? Isolates from clusters 2 and 3 are according to the phylogenetic tree even more related then isolates from clusters 1 and 2.
Thank you, we missed cluster 3 from the text.
- Row 328 – 330: the reference 38 should be at the end of the sentence “For example, a study in Iran assessed…” Ref 38 should not be at the end of the sentence with the results from your study.
Thank you, changed!
- Row 330 - 331: “Our study has identified 3 main clusters with a similarity between 80
and 99%, but very few isolates were completely identical.”
Row 233-234: “Representation of dendrogram revealed a similarity of 70% of cluster 1 and 2, indicating a possible ancestral relationship from the isolates”
Abstract: The molecular fingerprinting showed three Pae clusters, each including strains with 81%-99% similarity.
At which % of the similarity were the 3 clusters created? 70 or 80?
Also, according to the dendrogram, 100% identical isolates were not present in this study. Did you gain the same ERIC patterns for at least 2 isolates? According to the dendrogram, the most similar ones were isolates with 99% similarity – are you considering them as identical? In this case, you should write a sentence in the methods, that you are considering the isolates as identical with 98 – 99% similarity. – this note also refers to point 21.
Thank for noticing and making this to become clear. We included the information in a phrase: “As the threshold for potentially similar strains was set at 80%, by analyzing the dendrogram results, a number of 3 clusters were defined, represented in Figure 3, indicating a possible ancestral relationship from the isolates. Completely identical ERIC-PCR patterns were not identified within the studied isolates; nevertheless, similarity >98% was observed in few cases.”
- Row 333 – 336: “For example, Syrmis et al.” Why is at the end of this sentence reference
37? The results are from the reference 39. Also check the similarity threshold in ref 39.
Thank you, we rechecked and indeed was 85%
- Row 336: “Another review..” The previous study was an original article, I would just write “A review from 2018.”
OK, changed.
- Row 337 – 338: “.also in cystic fibrosis patients, and showed a heterogeneity of isolates.”
The reference 40 (Parkins et al. 2018)
„Whereas it was long presumed that each patient independently acquired unique strains of Pseudomonas aeruginosa present in their living environment, multiple studies have since demonstrated that shared strains of P. aeruginosa exist among individuals with CF. Many of these shared strains, often referred to as clonal or epidemic strains, can be transmitted from one CF individual to another, potentially reaching epidemic status“.
I recommend rewriting the whole sentence.
Thank you, we rewrote the sentence as follows: “A review from 2018 assessed the global prevalence of P. aeruginosa clonality by ERIC-PCR, PFGE or MLST, but in cystic fibrosis patients; the data was reported by a limited number of countries, but nevertheless showed the sharing of P. aeruginosa strains among individuals with this specific pathology”
- Row 343: “Identical genetic patterns are described to be associated with clonal distribution of Pae” Rewrite the sentence.
Thank you, we rewrote the phrase: “Identical genetic patterns are suggestive for clonal distribution of bacteria and infection outbreaks. This was presented for example by Opperman et al. in 2022, who found that the ST303 clone was spreading throughout the city”
- Row 346 - 347: “Another outbreak was proved by molecular methods by Bertrand et al., in 2000 in France, when 55 identical carbapenem-resistant Pae were isolated in a 1-month time”. Reference 43 (Bertrand et al. 2000)
“During a 30-month survey, 55 patients were colonized or infected by a single clone of Pseudomonas aeruginosa in a surgical intensive care unit (ICU)”.
Thank you for noticing this, we updated the text.
- Row 348 – 349: “The clonal spreading is not our case, as the ERIC-PCR patterns were, with few exceptions, not identical.” See point 15.
Thank you, rechecked!
- Some of the references are older then 10 years (e.g. Bertrand 2000) I recommend providing references no older than 5 years.
Thank you for the recommendation! Some of the references describe the situation in the past, wishing to reveal better the evolution of the resistance. In some situations, there are defining some of the terms, described for the first time in early 2000. When it comes to the example that you give us “Bertrand 2000”, there are only few studies describing the epidemiology implication of P. aeruginosa and the description of the genetic ERIC PCR.
- Names of bacteria are generally written in italics, see line 125.
We have rechecked this aspect throughout the manuscript.
- Figure 2 shows the wrong name of the antibiotic, the correct one is imipenem, not imipemenm.
Thank you for noticing this!

Reviewer 3 Report
This study focuses on the changes in antimicrobial resistance of Pseudomonas aeruginosa during the six years before and after the COVID-19 epidemic and analyzes the genetic similarity of isolated strains. The results shows that dermatology is the most important source of Pseudomonas aeruginosa in the tertiary hospital from Romania, the number of Pseudomonas aeruginosa detections shows a significant decline in 2020, and antimicrobial resistance does not change much. The study is interesting, but the manuscript needs to be polished. The following points should be addressed:
(1)Some of the figures and tables could be further optimized for focus. The decrease in the number of Pseudomonas aeruginosa is due to a reduced number of bacteriological tests, therefore the use of bar graphs in figure2 to compare the number of resistant/susceptible strains is not intuitive. It is recommended to use the R% data and plot the results for the 6 drugs in one heat map.
(2)What is the prevention and control policy in Romania during the continuation of COVID-19? What are the possible effects of these measures, besides affecting the number of Pseudomonas aeruginosa detections and antimicrobial resistance?
(3)Colistin is an important antibiotics in bacterial therapy, line 208 mentions that strains selected or ERIC-PCR has variable susceptibility to colistin, but the specific resistance data is not shown.
Author Response
We ensure you of our appreciation for the time spend trying to help us improve the quality of the manuscript! The reviewers’ comments were highly constructive and we are totally agreeing that they helped improving the manuscript. Hoping that our answers will respond to your expectations, we are addressing your comments point-by-point.
Reviewer #3
- Some of the figures and tables could be further optimized for focus. The decrease in the number of Pseudomonas aeruginosais due to a reducednumber of bacteriological tests, therefore the use of bar graphs in figure2 to compare the number of resistant/susceptible strains is not intuitive. It is recommended to use the R% data and plot the results for the 6 drugs in one heat map.
Thank you for the recommendation, the results were presented as you suggested on heatmap
- What is the prevention and control policy in Romania during the continuation of COVID-19? What are the possible effects of these measures, besides affecting the number of Pseudomonas aeruginosadetections and antimicrobial resistance?
Thank you, we discussed this as follows: “In Romania there are policies regarding the surveillance, prevention and limitation of infections associated to the medical assistance in health facilities since 2016, which guides the measures to be taken to limit this phenomenon. Despite this, the antibiotic resistance rate was and is still high in Romania. Nevertheless, our data show that in the last years, probably under the pressure of COVID-19 pandemics that forced the medical professionals to better comply with the antibiotic surveillance, or due to dissemination of antibiotic therapy guides, the resistance patterns changed to a more favorable situation. This is also shown by the European Centre for Disease Prevention and Control [48], as the percentage of carbapenem- or fluoroquinolone-resistant P. aeruginosa was lower in 2020-2021 compared with 2018-2019.”
- Colistin is an important antibiotics in bacterial therapy, line 208 mentions that strains selected or ERIC-PCR has variable susceptibility to colistin, but the specific resistance data is not shown.
According to EUCAST, colistin has to be tested by microdilution method. The Vitek or disk-diffusion methods that we used are not acceptable anymore according to the new standards (even if we screened this by the previous standards). Thus, we choose not to include anymore this information in the article.

Round 2
Reviewer 1 Report
Figure 2. Putting the values of the % similarity scale
Author Response
Figure 2. Putting the values of the % similarity scale
Thank you, indeed the numbers were to small to be seen, GelJ dendrogram created a small font, we changed the dimension on the font and now are more visible.
Reviewer 2 Report
First of all, I thank the authors of the manuscript for editing the text based on my comments.
The text has been adequately edited and I am now pleased to recommend the manuscript for acceptance.
Author Response
Thank you for your comments!